Investigating the molecular basis for heterophylly in the aquatic plant Potamogeton octandrus (Potamogetonaceae) with comparative transcriptomics

He Dingxuan 1
Guo Pin 2
Gugger Paul F. 3
Guo Youhao 1
Liu Xing xingliu@whu.edu.cn 1
Chen Jinming jmchen@wbgcas.cn 1192194647@qq.com 4
1 Laboratory of Plant Systematics and Evolutionary Biology, College of life Sciences, Wuhan University , Wuhan , Hubei , China
2 Department of Biological Sciences, Graduate School of Science, The University of Tokyo , Tokyo , Japan
3 Appalachian Laboratory, University of Maryland Center for Environmental Science , Frostburg , MD , USA
4 Key Laboratory of Aquatic Botany and Watershed Ecology, Wuhan Botanical Garden, Chinese Academy of Sciences , Wuhan , China
McCormick Sheila
Electronic publication date: 2018 Feb 28
Publication date: 2018
Volume: 6
Electronic Location ID: e4448
Received 2017 Oct 9; Accepted 2018 Feb 13
Copyright: ©2018 He et al.
Copyright year: 2018
Copyright holder: He et al.
License: This is an open access article distributed under the terms of the Creative Commons Attribution License, which permits unrestricted use, distribution, reproduction and adaptation in any medium and for any purpose provided that it is properly attributed. For attribution, the original author(s), title, publication source (PeerJ) and either DOI or URL of the article must be cited.
License URL: https://creativecommons.org/licenses/by/4.0/

Keywords: Gene expression, Potamogeton octandrus, Transcriptome, Heterophyllous leaves

Funding: Doctoral Program of Higher Education 20110141110019 National Natural Science Foundation of China 31270278 31570220 Strategic Priority Research Program of the Chinese Academy of Sciences XDPB02 Wuhan Botanical Garden, Chinese Academy of Sciences Y655261W03 This study was supported by the Specialized Research Fund for the Doctoral Program of Higher Education (20110141110019), the National Natural Science Foundation of China (Nos. 31270278 and 31570220), the Strategic Priority Research Program of the Chinese Academy of Sciences (No. XDPB02), and the Wuhan Botanical Garden, Chinese Academy of Sciences (No. Y655261W03). The funders had no role in study design, data collection and analysis, decision to publish, or preparation of the manuscript.

==============================
Many plant species exhibit different leaf morphologies within a single plant, or heterophylly. The molecular mechanisms regulating this phenomenon, however, have remained elusive. In this study, the transcriptomes of submerged and floating leaves of an aquatic heterophyllous plant, Potamogeton octandrus Poir, at different stages of development, were sequenced using high-throughput sequencing (RNA-Seq), in order to aid gene discovery and functional studies of genes involved in heterophylly. A total of 81,103 unigenes were identified in submerged and floating leaves and 6,822 differentially expressed genes (DEGs) were identified by comparing samples at differing time points of development. KEGG pathway enrichment analysis categorized these unigenes into 128 pathways. A total of 24,025 differentially expressed genes were involved in carbon metabolic pathways, biosynthesis of amino acids, ribosomal processes, and plant-pathogen interactions. In particular, KEGG pathway enrichment analysis categorized a total of 70 DEGs into plant hormone signal transduction pathways. The high-throughput transcriptomic results presented here highlight the potential for understanding the molecular mechanisms underlying heterophylly, which is still poorly understood. Further, these data provide a framework to better understand heterophyllous leaf development in P. octandrus via targeted studies utilizing gene cloning and functional analyses.

Introduction

The presence of two or more leaf forms on a single plant, or heterophylly, is widely observed across diverse plant species, and most notably in aquatic plants (Minorsky, 2003; Zotz, Wilhelm & Becker, 2011). In some cases, heterophylly is believed to be an adaptive response to the environment, and it has been linked to an increase in fitness (Cook & Johnson, 1968; Wells & Pigliucci, 2000; Minorsky, 2003). For example, the heterophylly of aquatic plants may increase their fitness by decreasing leaf damage, decreasing water loss, enhancing photosynthesis, or promoting sexual reproductive success (Winn, 1999a; Winn, 1999b; Wells & Pigliucci, 2000; Minorsky, 2003; Zhang et al., 2009; Zotz, Wilhelm & Becker, 2011). Accordingly, heterophylly has been used as a model system for studying gene-environment interactions (Pigliucci, 2010; Nakayama et al., 2014).

In the past century, numerous studies have described morphological changes in heterophyllous plants in response to environmental factors such as CO2 concentration, oxygen capacity, salt concentration, temperature, water levels, seasonal changes, and light intensity and quality (McCallum, 1902; Arber, 1920; Fassett, 1930; Sculthorpe, 1967; Cook & Johnson, 1968; Bodkin, Spence & Weeks, 1980; Deschamp & Cooke, 1984; Titus & Sullivan, 2001). More recent studies have also revealed that plant hormones, including ethylene, abscisic acid (ABA), and gibberellin (GA) can affect heterophyllous leaf formation in many plant species, including Potamogeton nodosus (Anderson, 1978), Hippuris vulgaris (Kane & Albert, 1987), Marsilea quadrifolia (Liu, 1984), Callitriche heterophylla (Deschamp & Cooke, 1985), Ranunculus flaellaris (Young & Horton, 1985; Young, Dengler & Horton, 1987), Ludwigia arcuata (Kuwabara et al., 2003), and Rorippa aquatica (Nakayama et al., 2014). For example, ethylene gas has been shown to induce the development of submerged-type leaves on terrestrial shoots of L. arcuate. ABA, however, induces the formation of terrestrial-type leaves on submerged shoots (Kuwabara et al., 2003). However, the molecular mechanisms regulating the alterations of leaf forms in heterophyllous plant species remain largely unclear.

Plant leaves arise from a group of initial cells termed shoot apical meristems (SAMs). The molecular mechanisms responsible for the initiation and maintenance of SAMs and polar processes of leaf expansion have been recently studied in detail (Hay & Tsiantis, 2006; Tsukaya, 2006; Uchida et al., 2007; Uchida et al., 2010; Shani et al., 2010; Moon & Hake, 2011). For example, SAMs are characterized by expression of the Class I KNOTTED1-LIKE HOMEOBOS (KNOX) gene (Smith et al., 1992; Jackson, Veit & Hake, 1994), and down-regulation of the KNOX gene in regions where the leaf primordia will initiate is one of the earliest indications of leaf development (Moon & Hake, 2011). However, few studies have investigated the molecular biological changes that occur during the course of heterophyllous transitions. Hsu et al. (2001) identified several early-response ABA-regulated genes, designated ABRH (ABA-responsive heterophylly), in the aquatic fern M. quadrifolia. ABRH genes encode transcription factors, protein kinases, membrane transporters, metabolic enzymes, and structural proteins. In addition, Chen et al. (2011) demonstrated that 9-cis-epoxycarotenoid dioxygenase 3 (NCED3) plays a key role in regulating ABA-mediated heterophylly via endogenous ABA in two different lily varieties. Nakayama et al. (2014) also showed that regulation of GA levels via KNOX1 genes is primarily responsible for modulating heterophylly in R. aquatica. Heterophylly occurs across diverse taxa and may have evolved through convergent evolution (Minorsky, 2003), and thus different developmental processes and molecular mechanisms may exist among different species. Further, heterophylly within a single plant is controlled through multiple signalling pathways (Lin & Yang, 1999; Hsu et al., 2001). Thus, additional study systems in diverse heterophyllous plants and the use of large comparative datasets generated at the whole genomic or transcriptomic levels would aid in elucidating the complex molecular mechanisms that regulate heterophylly.

Here, we investigated the molecular mechanisms underlying heterophylly in Potamogeton (Potamogetonaceae), which is a genus that comprises more than eight heterophyllous species (Guo et al., 2010). P. octandrus Poir (2n = 28), a heterophyllous pondweed with submerged and floating leaf forms, is a perennial aquatic herb that is self-compatible and can reproduce vegetatively through rhizomes, or sexually by selfing and outcrossing seeds. P. octandrus produces many floating leaves that are flat and ovate with sharp leaf tips and submerged leaves that are linear in shape (Fig. 1). During the initial development stage, all seedlings are submerged under water and the stem apex can produce sessile, linear, and entire submerged leaves arranged in a decussate phyllotaxy (Fig. 1A). When submerged stems reach the surface of the water, the stems begin plagiotropic growth, and the stem apex can produce both floating and submerged leaves as the stem elongates (Fig. 1B).

Figure 1 Morphological features of P. octandrus.

(A) The initial developmental stage of the plant that produces only submerged leaves. (B) The later developmental stage of the plant that produces both floating and submerged leaves.

The transcriptomes of submerged and floating P. octandrus leaves at different time points along the developmental trajectory were sequenced using the Illumina RNA-Seq platform. De novo assembly of the DNA reads generated from submerged and floating leaf materials were then used as the reference transcriptome, and short sequence reads that were generated at several time points within the development of submerged and floating leaves were mapped to the assembled transcriptome in order to identify genes exhibiting differential expression between leaf morphologies. These results provide a crucial reference transcriptome for investigating the regulatory mechanisms underlying each leaf form, in addition to a list of candidate genes that are likely to be involved in the transition to heterophylly. Further, these results add to the growing body of data aimed at understanding the regulatory mechanisms of heterophylly among heterophyllous plants.

Materials and Methods

Plant materials

A P. octandrus plant was collected from a Tongcheng population (29°16′05.6″N, 113°48′46.9″E) in the Hubei Province, China, and used for transcriptomic analysis. The plant was transplanted to a greenhouse at Wuhan University in April 2015 and the seeds were harvested in August 2015. Seedlings were cultured in a pool at Wuhan University and floating and submerged leaf forms were sampled at the following time points: (1) floating leaves with length less than 0.5 cm (JFL; 72 h); (2) floating leaves with length more than 1 cm (AFL; 84 h); (3) submerged leaves with length less than 1.5 cm (JSL; 96 h); (4) submerged leaves with length more than 3 cm (ASL; 120 h), in addition to (5) leaf-shoots of plants with stems that have an apex just reaching the water surface (shoot; 24 h). In the shoot category, identification of which leaf form the shoot will develop into is not yet possible. The “shoot” was considered as the initial stage of either floating or submerged leaves. Fifteen samples with three biological replicates for each leaf/shoot form were collected in total. Sampled tissues were immediately frozen in liquid nitrogen and stored at −80°C until further analysis.

RNA extraction, cDNA library construction and sequencing

Total RNA from each sample was extracted using the TRIzol reagent (Invitrogen, Carlsbad, CA, USA) according to the manufacturer’s instructions. RNA was then treated with RNase-free DNase I (Fermentas, ThermoFisher, Waltham, MA, USA). The quantity and quality of RNA was assessed using 1% agarose gel electrophoresis of extracted RNA, in addition to analyses with an RNA 6000 Nano Assay Kit, and an Agilent 2100 Bioanalyzer (Agilent Technologies, Palo Alto, CA, USA). Total extracted RNA was stored at −80°C.

cDNA libraries were then assembled for the 15 RNA samples (T01-T15). cDNA was synthesized using a SuperScript VILO cDNA Synthesis Kit (Invitrogen., Carlsbad, CA, USA) according to the manufacturer’s protocol. A total of 10 µg of RNA was purified using oligo (dT) magnetic beads in order to enrich for poly (A) mRNA. Fragmentation buffer was then added to split mRNA into short fragments. Fragments were used as templates to synthesize first-strand cDNA using random hexamer-primers (Invitrogen, Carlsbad, CA, USA) and 1000 U of SuperScriptTM II reverse transcriptase (Invitrogen, Carlsbad, CA, USA). RNase H and DNA polymerase I were used to synthesize second-strand cDNA. The short fragments were then amended with adapters and end repair ligation. The products were first purified using a QIAquick® Polymerase Chain Reaction (PCR) Extraction Kit (Qiagen, Hilden, Germany), and then enriched with PCR (15 cycles) to create the final cDNA libraries. The 15 cDNA libraries were then sequenced to obtain 150 bp paired-end reads using the Illumina HiSeq™ 2500 sequencing platform (Illumina Inc., San Diego, CA, USA). The raw sequencing data were deposited into the NCBI Short Reads Archive (SRA) under the accession numbers SRR6649465, SRR6649466, SRR6649467, SRR6649858, SRR6649859, SRR6649860, SRR6649878, SRR6649879, SRR6649880, SRR6650048, SRR6650049, SRR6650050, SRR6655609, SRR6655610, SRR6655611.

Sequence assembly and gene annotation

The raw reads were cleaned by removing low quality reads, adapter reads, and reads with 5% or more unknown nucleotides. Transcriptome de novo assembly was performed with the clean reads that resulted from all 15 samples using the Trinity program (Grabherr et al., 2011) with the min_kmer_cov parameter set to 2 by default. Overlapping sequence reads were first combined to form contigs without gaps, and reads were then reverse mapped to the contigs. Paired-end sequencing allowed the inference of contigs arising from the same transcript, and distances between these contigs were determined. Contigs were connected in Trinity and sequences that could not be extended on either end were identified, followed by construction of ‘unigenes’ that represented transcripts from the same locus. The completeness of the P. octandrus transcriptome assembly was estimated using reciprocal BLAST searches against annotated proteins from the genome of a seagrass species closely related to P. octandrus, Zostera marina (Olsen et al., 2016) and an E-value threshold of 1e−5.

The putative functions of unigenes were then assessed using BLASTx (E-value ≤10−5) and several protein databases including the: NCBI non-redundant protein (nr), Swiss-Prot, Clusters of Orthologous Groups (COG), euKaryotic Orthologous Groups (KOG), eggNOG, Protein family (Pfam), Gene Ontology (GO), and Kyoto Encyclopedia of Genes and Genomes (KEGG) databases. The sequencing directions of unigenes were determined based on the best alignments. The ESTScan program (version 3.0.1, http://www.ch.embnet.org/software/ESTScan.html) was used to determine sequencing direction when a unigene could not be aligned with any of the above databases. Blast2GO 2.5.0 (Conesa et al., 2005) was used to compare and determine unigene Gene Ontology (GO) annotations. GO functional classifications for all annotated genes were obtained with the program WEGO (Ye et al., 2006), which was also used to plot the distribution of gene functions.

Analysis of differentially expressed genes (DEGs)

To quantify the abundance of transcripts, clean reads from each sample were then mapped onto the assembled reference transcriptome using the Bowtie2 read-mapping, alignment software (Langmead & Salzberg, 2012). Gene expression levels and comparisons of gene fragments per kb per million (FPKM) fragments values (Marioni et al., 2008) were then assessed using likelihood ratio tests in order to statistically identify differentially expressed genes (DEGs). A previously developed algorithm (Audic & Claverie, 1997) was used to assess DEGs between samples from different leaf form stages (control/experimental: shoot/JFL, shoot/AFL, JFL/AFL, shoot/JSL, shoot/ASL, JSL/ASL). False discovery rate (FDR) control was used to correct for P values for multiple hypotheses. Genes with changes in expression that were two-fold or greater and had FDR values ≤ 0.01 were identified as differentially expressed and then absolute values of log2 (foldchange) ≥ 1 were regarded as DEGs.

GO functional analysis was used to assess GO functional enrichment and GO functional annotation for the DEGs. All DEGs were mapped to each term of the GO database. We then calculated the gene abundances for each GO term. A hypergeometric test was then applied to identify significantly enriched GO terms in DEGs relative to the genome background. P-values from the GO enrichment analysis were adjusted using Bonferroni’s correction. A corrected P-value ≤ 0.05 was selected as the threshold for significantly enriched GO terms. DEGs were then mapped to terms in the KEGG database to obtain enriched pathway terms (Kanehisa & Goto, 2000). Pathways with an FDR value of ≤ 0.05 were considered significant.

Quantitative real-time PCR (qPCR) analysis

To validate the transcriptomic data, 16 DEGs were randomly selected and their expression profiles were investigated by qPCR. Total RNA was extracted from P. octandrus leaves in the five groups and cDNA libraries were constructed as described above. A SYBR Premix Ex Taq™ Kit (TaKaRa) was used for the qPCR reactions in a BioRad Real-Time thermal cycler system (BioRad, Hercules, CA, USA). Gene-specific primers were designed using Primer Premier5, and the primer sequences are provided in File S5. The PCR cycling regimen was developed according to the manufacturer’s protocol and consisted of 95°C for 30 s, 40 cycles of 95°C for 5 s and 58°C for 30 s. Each reaction was performed in triplicate and β-actin was used as a reference gene. The 2−ΔΔCt method was used to calculate relative gene expression levels (Livak & Schmittgen, 2001).

Results

Illumina sequencing and de novo assembly

After removal of duplicate sequences, adaptor sequences, low quality reads, and reads with ambiguous bases, a total of 135.83 Gbp clean sequence data with Q30 ≥ 96.0% were obtained from the 15 cDNA libraries. Clean reads were pooled and de novo assembled into 81,103 unigenes, which had an average length of 841 bp (N50 = 1,713 bp) (Table 1). The most abundant length fraction of Unigenes were 201–300 bp, followed by 500–1,000 bp and 300–500 bp sequences (Table 1). Cleaned reads were then mapped to the unigenes to assess sequencing randomness, which was found to be sufficient (File S1). Further read mapping statistics for each sample are provided in File S2. About 91.1% (18,738/20,559) of all annotated proteins from a closely related seagrass species could be identified in the P. octandrus transcriptome, indicating that our transcriptome assembly likely covers the majority of the core P. octandrus transcriptome. In contrast, only 60.5% (101,218/ 167,270) of P. octandrus transcripts could be annotated using the closely related seagrass species proteins, suggesting that the remainder of the unannotated transcripts are from lineage-specific genes or long non-coding RNAs.

Table 1 Overview of the sequencing and assembly from RNA-Seq data.

Length	Transcript	Unigene	
200–300	40,102(23.97%)	38,142(44.30%)	
300–500	17,132(10.24%)	11,774(13.67%)	
500–1,000	30,450(18.20%)	15,041(17.47%)	
1,000–2,000	39,384(23.55%)	11,399(13.24%)	
2,000+	40,202(24.03%)	9,747(11.32%)	
Total number	167,270	86,103	
Total length	224,870,533	72,422,038	
N50 length	2,273	1,713	
Mean length	1,344.36	841.11	

Functional annotation and classification of unigenes

A total of 48,235 (56% of all unigenes) consensus sequences were annotated using BLASTx (E-value ≤1 × 10−5) against several public protein databases including NCBI-nr, Swiss-Prot, COG, KOG, eggNOG, Pfam, GO, and KEGG (File S3). Among the annotated unigenes, 24,025 were categorized into 52 functional subcategories under three main GO categories: cellular components, molecular functions, and biological processes (Fig. 2). The GO category “biological processes” represented the largest number of GO annotations, with metabolic processes, cellular processes, and single-organism processes comprising the three most abundant top-level subcategories. In the cellular components category, the “cell” and “cell part” categories were most represented, while in the molecular functions category, the “catalytic activity” subgroup was prominent, followed by the “binding” group. In the COG and KOG functional classifications, “translation, ribosomal structure and biogenesis” associated proteins represented the largest group (Files S4 and S5). Lastly, a total of 22,346 unigenes were assigned to 128 KEGG pathways.

Figure 2 Gene ontology (GO) annotations of all detected genes.

Three main categories including “biological process”, “cellular component”, and “molecular function” were summarized.

DEG identification and functional analysis

A total of 6,822 differentially expressed genes (DEGs) were identified by comparing samples from each leaf form group (control/experiment: shoot/JFL, shoot/AFL, JFL/AFL, shoot/JSL, shoot/ASL, JSL/ASL; File S6). A Venn diagram was used to visualize the numbers of overlapping and unique DEGs among and within groups (Figs. 3A and 3B). The shoot/JFL comparison contained the most DEGs (3,706), whereas the smallest number of DEGs (64) was detected in the JSL/ASL comparison. Among the DEGs associated with the development of floating leaves, 2,186, 1,621 and 78 were up-regulated, while 1,835, 1,679 and 332 were down-regulated in comparisons of shoot/JFL, shoot/AFL, and JFL/AFL, respectively. During the development of submerged leaves, 2,413, 2,242 and 45 DEGs were found to be up-regulated, while 1,458, 1,284 and 33 were down-regulated in the shoot/JSL, shoot/ASL, and JSL/ASL comparisons, respectively (Fig. 3C). These results indicated that the greatest differential expression occurred in the early stages of development for both floating and submerged leaves (shoot/JFL and shoot/JSL). Further, the number of up-regulated DEGs was higher than the number of down-regulated DEGs during the development of submerged leaves. However, the number of up-regulated and down-regulated DEGs was essentially equivalent during the development of floating leaves.

Figure 3 DEGs identified in comparisons among different P. octandrus leaf development stages.

“T01, T04, T12”, “T02, T03, T05”, “T06, T08, T13”, “T07, T10, T14”, and “T08, T11, T15” indicate three biological replicates of shoot, juvenile floating leaves, adult floating leaves, juvenile submerged leaves, and adult submerged leaves, respectively. (A) Venn diagram showing common and unique DEGs among different comparisons of floating leaves. (B) Venn diagram showing common and unique DEGs among different comparisons of submerged leaves. (C) Expression patterns of DEGs among different comparisons. The numbers above each bar indicate the total number of genes in each group.

The number of DEGs that were annotated in the shoot/JSL and shoot/ASL comparisons during floating leaf development was greater than those in the JSL/ASL comparison (Fig. 4). A similar pattern was found in comparisons for the development of submerged leaves (Fig. 5). The highest number of DEGs for each comparison was identified in the “biological process” category, while the most abundant sub-group terms were in the “cellular process”, “metabolic process” and “single-organism” categories for the different groups of submerged and floating leaves. Enriched GO terms during the earlier time points of leaf development (JFL and JSL) included “translation”, “regulation of transcription, DNA-templated”, “RNA methylation”, and “photosynthesis”. However, the GO terms enriched during the later developmental time points were assigned to “response to salt stress” (AFL and ASL), “salicylic acid biosynthetic process” (AFL), and “negative regulation of programmed cell death” (ASL). In the cellular component category, “cell” and “cell part” were the two most highly represented components throughout the developmental time points for all leaves. The GO terms “ribosome” and “cytosolic small ribosomal subunit” were enriched in different stages of leaf development, while the GO terms “cytoplasmic membrane-bounded vesicle” (JFL) and “chloroplast envelope” (JSL) were enriched in the earlier developmental time points. In the molecular function category, DEGs that mapped to “catalytic activity” and “binding” were present in high proportions through all developmental time points. The results of GO enrichment analysis are presented in File S7.

Figure 4 GO functional classifications of DEGs identified from comparisons among different groups during floating leaf development.

“T01, T04, T12”, “T02, T03, T05”, and “T06, T08, T13” indicate three biological replicates of shoot, juvenile floating leaves, and adult floating leaves, respectively.

Figure 5 GO functional classifications of DEGs identified from comparisons among different groups during submerged leaf development.

“T01, T04, T12”, “T07, T10, T14”, and “T08, T11, T15” indicate three biological replicates of shoot, juvenile submerged leaves, and adult submerged leaves, respectively.

Annotated sequences were then searched against the KEGG pathways database. Among the annotated DEGs, 1,490, 1,141, 135, 1,425, 1,362, and 30 were assigned to 114, 114, 57, 114, 111, and seven pathways in the shoot/JFL, shoot/AFL, JFL/AFL, shoot/JSL, shoot/ASL, and JSL/ASL comparisons, respectively (File S8). The pathways with the largest proportions of DEGs were “ribosome”, “biosynthesis of amino acids”, “carbon metabolism”, and “plant-pathogen interaction” in the shoot/JFL, shoot/AFL, shoot/JSL, and shoot/ASL comparisons, respectively (File S8). However, these four pathways accounted for a lower percentage of the proportion in the JFL/AFL and JSL/ASL groups, which may be important in the early stages of development of floating and submerged leaves. “Plant hormone signal transduction” was the most enriched cluster among the DEGs, indicating that proteins in this category may be important in the morphological and physiological differentiation that occurs in the early stages of leaf development.

Analysis of transcription factors associated with heterophyllous leaf types

To assess the function of transcription factors (TFs) during the development of heterophyllous leaves, we identified P. octandrus TF genes using the Plant Transcription Factor Database (http://planttfdb.cbi.pku.edu.cn/). In total, we identified 1,681 putative TF genes that were categorized into 48 families. Four-hundred and sixty-nine of the 1,681 putative TF genes that were from 42 families showed significant differential expression between the leaf groups (Fig. 6 and File S9). Most of these genes (53%) were expressed at the highest levels in the AFL, ASL, JFL, and JSL groups (Group 2; G2). Only 37% were expressed at the highest levels in the “shoot” group (Group 1; G1). Lastly, an additional 10% exhibited highest expression in the JFL group (Group 3; G3). We further investigated differential expression at the family-specific level (Fig. 7). The WRKY, C3H, and AP2 families of TFs, which play roles in various processes including stress response, leaf senescence, and leaf epidermal cell identity, were highly expressed in G2 (File S10). Many TFs that mediate hormone signaling, including ethylene signaling (ERF family) and auxin signaling (NAC family), were highly expressed during the development of floating and submerged leaves (G2). The HD-Zip family members that have been proposed as regulators of vascular development, stomatal complex morphogenesis, leaf polarity, and epidermal cell differentiation, were most highly expressed in G2. MYB, ARF, and B3 TFs are involved in light and hormone signaling pathways. These genes were most expressed in the “shoot” (G1) group, where cell differentiation and cellular morphogenesis are modulated. Homeobox genes, which participate in a number of developmental events, were also highly expressed in G1. Several FAR1 and bHLH TFs, which have been reported to function in light signaling and stomatal development, were enriched in the earlier time point floating leaves (G3). These results suggest that transcriptional regulatory genes are critical for a wide variety of developmental processes in the leaf transcriptome.

Figure 6 Dendrogram showing similarities in transcription factor expression profiles among samples.

Three transcription factor cluster groups (G1, G2 and G3) resulted from the 469 significantly differentially expressed transcription factors in leaves from different stages of development.

Figure 7 Distribution of transcription factor families among the three transcription factor cluster groups (G1, G2 and G3).

Hormone metabolism and signalling pathways among heterophyllous leaf samples

KEGG annotation revealed that most DEGs were classified in the “plant hormone signal transduction” category. A total of 70 DEGs were annotated as diverse hormone-related genes, including those involved in signalling with abscisic acid (ABA), auxin, cytokinin (CTK), ethylene, jasmonic acid (JA), gibberellin (GA), brassinosteroid, and salicylic acid (SA).

Among hormone signaling genes, most were involved in auxin response pathways during the development of both floating and submerged leaves. Genes encoding the auxin response factor (ARF) were down-regulated, while genes encoding SAUR family members were differentially expressed, with two DEGs up-regulated and two down-regulated. Genes encoding an auxin influx transport protein (AUX1) and an auxin-responsive protein AUX/IAA were up-regulated in floating leaves, whereas those genes were down-regulated in submerged leaves.

Eight genes that were associated with ABA and GA pathways were highly expressed in floating leaves, including those encoding PYR (pyrabactin resistance)/PYL (PYR1-like), SNF1 related protein kinase 2 (SnRK2), ABA responsive element binding factor (ABF), and phytochrome-interacting factor (PIF). Genes that were related to GA and ethylene pathways were also highly expressed in submerged leaves, including one that encoded gibberellin receptor (GID1) and two encoding the ethylene-responsive transcription factor 1 (ERF1).

Among the 31 hormone-related genes that were up-regulated during the differing developmental time points of floating leaves, four were involved in the ABA signal transduction pathway, four were associated with the GA signal transduction pathway, five were responsive to auxin stimulation, three were associated with ethylene-mediated signaling, two were responsive to CTK stimulus, and three were related to brassinosteroid signaling. Most of these genes were significantly up-regulated in the earlier time point of floating leaves. Additionally, 34 genes involved in the hormone signal transduction pathway were up-regulated during the different developmental points of submerged leaves. When compared with floating leaves, more genes were involved in GA, ethylene and CTK signal transduction pathways during the development of submerged leaves (File S11).

Expression patterns of ABA- and GA-biosynthesis genes

In the ABA biosynthesis pathway, seven DEGs that were annotated to encode zeaxanthin epoxidase (ZEP), 9-cis-epoxycarotenoid dioxygenase (NCED), and abscisic-aldehyde oxidase (AAO), exhibited different levels of expression. Genes encoding ZEP and NCED were highly expressed during the different stages of submerged leaves and particularly highly expressed in floating leaves. Further, the expression of genes encoding AAO was down-regulated in the later stages of development of submerged leaves.

In the GA biosynthesis pathway, the expression of one DEG encoding gibberellin 20 oxidase (GA20ox), which is a key enzyme in GA biosynthesis, was down-regulated throughout developmental time points, except during the later time points of floating leaves. Additionally, two genes that encode gibberellin 2beta-dioxygenase (GA2ox), which has an inactive effect on GAs, were down-regulated. These results suggested that GA signaling pathways are important in early stages of leaf development.

Expression patterns of genes mapped to “stomatal complex morphogenesis” and “cuticle development”

Of the genes that were functionally annotated, some were associated with the functional groups “stomatal complex morphogenesis” and “cuticle development” throughout leaf development time points. Twenty-three DEGs were associated with “stomatal complex morphogenesis”, including 12 that were up-regulated during different time points of floating leaf development, and particularly highly expressed in the earlier time point for floating leaves. Three DEGs were also down-regulated in submerged leaves. The up-regulated DEGs were annotated as phosphoribulokinase, chloroplastic, auxin-binding protein ABP19a, leaf isozyme, and phototropins.

A total of 11 DEGs were mapped to “cuticle development”, four of which were up-regulated during the different time points of floating leaves and three that were down-regulated. The up-regulated DEGs include orthologs of 3-ketoacyl-CoA synthase 10 and beta-ketoacyl-CoA synthase like protein. The large number of up-regulated stomatal morphogensis and cuticle development genes may indicate regulation of leaf morphology and structure.

Validation of DEGs expression by qPCR

Sixteen candidate DEGs were selected for qPCR validation of expression levels. These genes were categorized into groups that were related to transcription factors, biological processes, and hormone signaling. Five transcription factors were tested, including NF-YB, MYB, GRF, NAC, and NF-YA (c260025.graph_c1, c265373.graph_c0, c267010.graph_c0, c268319.graph_c0, c269000.graph_c0). Additionally, six genes that were involved in biological processes were analyzed, including guard cell differentiation (c260078.graph_c1), stomatal complex morphogenesis (c260159.graph_c0, c262330.graph_c0, c267324.graph_c0), wax biosynthetic processes (c271037.graph_c1), and cuticle development (c271065.graph_c0). In hormone signaling, AUX/IAA (c264173.graph_c0), AUX1 (c264621.graph_c1), ARF (c268917.graph_c0), GA2ox (c266852.graph_c0), and GA20ox (c268628.graph_c0) were selected. The gene annotations of these candidate DEGs are provided in File S12. The correlation between RNA-Seq results (fold change) and qPCR results (2−ΔΔCT) was measured by plotting the log2 fold changes (File S13a). The results suggested that the qPCR abundances of genes were significant similar (r2 = 0.65, P < 0.01) to the RNA-Seq data (File S13b).

Discussion

A growing body of research has indicated that heterophylly is a common, adaptive response by plants to increase their fitness through responses to environmental changes. However, the molecular mechanisms underlying these processes have remained unclear. Comparative transcriptomic analysis is an efficient method for discovering genes and investigating biochemical pathways that are involved in physiological processes, and has recently become an increasingly tractable methodology (Varshney et al., 2009; Ozsolak & Milos, 2011; Shi et al., 2011; Mutasa-Goettgen et al., 2012; Yang et al., 2014). Here, leaves of the aquatic heterophyllous plant P. octandrus that has two leaf forms (submerged and floating), were subjected to transcriptomic sequencing to identify differentially expressed genes among the two leaf morphologies at differing time points of leaf development. Overall, 81,103 unigenes were assembled and 48,235 unigenes were annotated against public protein databases. A total of 6,822 differentially expressed genes (DEGs) were identified in comparisons between time points. KEGG pathway enrichment analysis was used to sort a number of DEGs into plant hormone signal transduction pathways, including Cytokinin, Auxin, Abscisic Acid, Gibberellin, Brassinosteroid, Ethylene, and Jasmonic Acid related pathways.

The initial stage of leaf shoot development is enriched in metabolic processes, cellular processes, and responses to stimuli. The expression levels of genes related to leaf morphogenesis, photomorphogenesis, and hormone signaling increased during developmental time points of submerged and floating leaves. This was especially evident in the younger floating leaves, where genes involved in establishing stomata and leaf petioles were more highly expressed than in other tissues. In older submerged leaves, genes associated with leaf senescence were more highly expressed than in other tissues. These gene transcriptional regulations were coincident with the developmental dynamics of heterophyllous leaves.

When grouped into KEGG pathways, most DEGs were associated with hormone signaling. Exogenous and endogenous ABA and GA can regulate leaf form alteration in heterophyllous plant species (Allsopp, 1962; Deschamp & Cooke, 1984; Gee & Anderson, 1998; Kuwabara et al., 2003). The ABA-mediated regulation of heterophyllous morphological changes has been studied intensively (Wanke, 2011; Nakayama et al., 2012). Here, a large number of hormone-related DEGs were associated with ABA and GA signal transduction pathways, suggesting that genes responsive to ABA and GA might play an important role in heterophyllous leaf formation in P. octandrus. The genes encoding the PYR/PYL ABA receptor, which interacts with PP2C phosphatases and is a component of the ABA signaling pathway, were up-regulated in floating leaves. Further, genes encoding SnRF2 proteins were also up-regulated. Activated SnRF2 proteins can phosphorylate downstream targets, such as AREB/ABF transcription factors and contribute to those signaling pathways (Cutler et al., 2010). Endogenous levels of ABA have been shown to increase in leaves of water-stressed terrestrial plants and ABA plays a pivotal role in drought stress in terrestrial plant species (Walton & Li, 1995). Thus, ABA may have a similar function in the initiation of heterophyllous leaves in response to the transition from submerged to aerial conditions in P. octandrus (Goliber & Feldman, 1989). Plant cells respond to environmental stimuli through a series of intracellular signals. To minimize transpirational water loss, ABA controls stomatal closure (Hirayama & Shinozaki, 2010). This change is induced by transcriptional reprogramming via the ABA signalosome complex (PYP/PYL-PP2C-SnRK2). With increasing concentrations of ABA, the ABA signaling complex (PYP/PYL-PP2C-SnRK2) can cause stomatal closure in guard cells in a calcium-independent manner and through key biochemical messengers (Geiger et al., 2011).

Many enzymes in ABA biosynthesis are also induced during the drought stress response, including the key enzymes ZEP, NCED, and AAO (Iuchi et al., 2001; Chen et al., 2011; Wanke, 2011). Our results indicated that 19 genes encoding ZEP, NCED, and AAO exhibited different expression levels among leaf morphological groups. Six of the 19 genes were highly expressed in floating leaves and up-regulated when compared to leaf shoots and submerged leaves, while only one gene was down-regulated. These results suggest that differential expression of genes that encode key enzymes in ABA biosynthesis might control ABA function at specific stages during the initiation of heterophyllous leaf changes. Thus, ABA may play a complex role in signalling transduction during heterophylly development in P. octandrus.

Recently, ethylene has been suggested to influence the formation of heterophyllous leaves in L. arcuata in a manner that is opposite of ABA, and also has been suggested to be an endogenous factor that induces the formation of submerged leaves (Kuwabara, Tsukaya & Nagata, 2001). Our results indicated that the expression of three genes encoding 1-aminocyclopropane-1-carboxylate synthase (ACS), a rate-limiting enzyme in ethylene biosynthesis, were highly expressed in floating leaves relative to leaf shoots and submerged leaves. GA likely has an antagonistic effect on heterophylly in aquatic plants and is also involved in the formation of submerged leaves. However, it induces heterophylly only indirectly through ethylene (Kuwabara, Tsukaya & Nagata, 2001). Therefore, changes in endogenous ABA concentrations can influence the formation of aerial leaves and feedback antagonistically on ethylene and GA (Wanke, 2011). Our results also indicated that two genes encoding GA2ox, which has an inactive effect on GA, were down-regulated in submerged leaves. Taken together, these findings suggest that ABA signalling was enhanced in the aerial leaves of heterophyllous plant species.

Transcription factors (TFs) that target promoter regions regulate the concentrations of proteins as limiting factors. They also play an important role in responses to environmental stress (Yuan & Perry, 2011) and plant development. Previous studies have demonstrated that many genes in the AP2/EREBP family participate in the transcriptional regulation of processes related to growth and development. For example, one member of the ERF gene family, ESR1, regulates shoot regeneration (Banno et al., 2001), while overexpression of SHN gene that are AP2/EREBP transcription factors, increases cuticular wax (Aharoni et al., 2004). Further, the Glossy15 gene from maize regulates leaf epidermal cell identity (Moose & Sisco, 1996), and the LEAFY PETIOLE (LEP) gene influences leaf petiole development in Arabidopsis thaliana (Van der Graaff et al., 2000). Here, we found that 48 TFs are predominantly expressed in P. octandrus and 42 TFs are up- or down-regulated among morphological comparisons. Among these TFs, several ERF homologs to SHN and LEAFY PETIOLE exhibited up-regulation at different time points of leaf development, which is consistent with thicker waxy cuticles and extended leaf petioles on floating leaves of P. octandrus. In addition, we detected a close homolog of maize Glossy15, which was highly expressed in floating and submerged leaves, suggesting a similar function of ERF in leaves of P. octandrus and maize.

The KNOX homeobox protein (Knotted1-like homebox, KNOX) is a homeodomain transcription factor that maintains cell pluripotency in plant shoot apical meristems (SAM) (Vollbrecht et al., 1991). Three KNOTTED1-like HOMEOBOX (KNOX) homologs were detected in our DEG analysis, and all of these were highly expressed in leaf shoots. KNOX proteins regulate the homeostasis of CTK and GA in order to maintain meristematic cells in an undifferentiated state (Shani, Yanai & Ori, 2006). CTK is a plant hormone involved in cell proliferation while GA controls leaf morphogenesis (Hooley, 1994; Mok & Mok, 2001). These results imply that regulation of GA levels by KNOX1 genes is involved in regulating heterophylly in P. octandrus.

bZIP TFs regulate a variety of plant development and abiotic resistance processes. For example, AtbZIP1 from Arabidopsis regulates ABA signal transduction by binding ABA-responsive elements (ABREs) and altering the expression of ABA-responsive genes (Sun et al., 2011). Here, three ABRE binding factors (AREB/ABF) were identified that were differentially expressed between leaf shoots, floating leaves and submerged leaves. These findings suggest that morphological differences between heterophyllous leaves may be directed by genes from multiple functional groups, such as bZIP genes.

We also identified several other TFs involved in ABA signalling. For example, the NAC TF, ANAC072, was detected and responds to exogenous ABA and may regulate ABA-responsive genes (Tran et al., 2004). Moreover, ABA regulates gene expression through additional TFs such as MYB, HD-ZF, B3 and bHLH (Fujita et al., 2011). Our data indicate that diverse TFs may be involved in heterophyllous leaf development and that the genes are both down-regulated and up-regulated, suggesting that TFs may be involved in different processes during heterophyllous leaf development.

Conclusions

Here, we describe the production, assembly, and annotation of transcriptomes from submerged and floating leaves of a heterophyllous aquatic plant, P. octandrus. We identified several genes that showed differential expression at different time points of development during the onset of heterophylly in this species. These results inform on a more complete understanding of the molecular mechanisms underlying heterophylly in plants. In particular, these results can be used for cloning and functional studies of genes that are involved in heterophylly development in order to further probe the mechanistic basis for this important developmental phenomenon.

Supplemental Information

Supplemental Information 1 Supplemental files

Click here for additional data file.

Additional Information and Declarations

Competing Interests

Author Contributions

Data Availability

The authors declare there are no competing interests.

Dingxuan He conceived and designed the experiments, performed the experiments, analyzed the data, contributed reagents/materials/analysis tools, prepared figures and/or tables, authored or reviewed drafts of the paper, approved the final draft.

Pin Guo conceived and designed the experiments, performed the experiments, analyzed the data, prepared figures and/or tables, authored or reviewed drafts of the paper, approved the final draft.

Paul F. Gugger analyzed the data, authored or reviewed drafts of the paper, approved the final draft.

Youhao Guo contributed reagents/materials/analysis tools, prepared figures and/or tables, authored or reviewed drafts of the paper, approved the final draft.

Xing Liu and Jinming Chen conceived and designed the experiments, authored or reviewed drafts of the paper, approved the final draft.

The following information was supplied regarding data availability:

The raw sequencing data were deposited into the NCBI Short Reads Archive (SRA) under the accession numbers SRR6649465, SRR6649466, SRR6649467, SRR6649858, SRR6649859, SRR6649860, SRR6649878, SRR6649879, SRR6649880, SRR6650048, SRR6650049, SRR6650050, SRR6655609, SRR6655610, SRR6655611.

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
