# Peer review of "Investigating the molecular basis for heterophylly in the aquatic plant Potamogeton octandrus (Potamogetonaceae) with comparative transcriptomics"

_PeerJ, doi:10.7717/peerj.4448_

## Round 0.1 · original submission · Major Revisions

Both reviewers found value in your work but also note significant changes required before submission. I agree with the critiques of the reviewers and ask that you address them all before resubmission.

·

Basic reporting

The paper entitled “Comparative transcriptomic analysis of heterophylly of the aquatic plant Potamogeton octandrus (Potamogetonaceae).” describes initial result of transcriptopme analysis to reveal the underlying mechanism of the heterophylly.

English is clear and professional enough. Background information and the aim of the research are well described.

Experimental design

The design of the research is reasonable and the experiments and analyses are carefully performed.

1) From line 246, the author mention Go enrichment analysis, but I do not find the data. Please show the results.

Validity of the findings

1) In this paper, authors identified DEGs by comparing each developmental stages, not by comparing between floating and submerged leaves. Thus, it is not clear whether these DEGs are involved in regulating the heterophylly. Some conclusion and discussion might be too speculative and misleading. If they want to discuss the mechanism of the heterophylly in this paper, author should conduct the comparison between floating and submerged leaves. Or authors can mention the 22 transcription factors in cluster G3 in figure 6, because their expression are clearly different between JFl and AFL.


2) Definition of “unigene” and unigene sequence is unclear. I think that unigene is same as “gene” output from Trinity in this paper. Gene defined by Trinity includes one or more transcript sequences. But one unigene sequence was used for annotation. If gene has 2 or more transcript isoform sequences, What does unigene sequence mean?

Additional comments

some minor comments,

1) through out the text, some gene names are not italicized
2) Line130: Agarose gel isn’t used for Bioanalyzer. Is either agarose electrophoresis or Bioanalyzer assay or both used?
3) Line 133 CDNA>cDNA
4) Line134: I can’t find cDNA Synthesis Kit (Illumina Inc., San Diego, CA, USA) to make RNAseq library. Is this product name correct?
5) Line 154 (NCBI............... > (NCBI............... )
6)Line186:Did you use the library kit for NGS library kit to make cDNA for qPCR?
7) From Additional file 6.xlsx, "#NAME?" should be removed.
8) I suggest showing sample name (shoot, JFL) rather than library ID in figure.

Reviewer 2 ·

Basic reporting

The manuscript is written in clear, understandable language.
The introduction is for the most part relevant, but is missing important details about the study system. Is anything known about the development of these different leaf types? Is this a novel innovation or found throughout the genus? What is the ploidy of this species? The authors also choose to describe the specific form of heterophylly in this species in the methods. This seems to be more appropriate material for the introduction. I think that the specific form of heterophylly is important to interpret the results (as is the biological function of these leaf types) but the authors treat heterophylly as a general process.
The structure of the manuscript is appropriate to the journal and the raw data is submitted to an appropriate database.
The figures are generally understandable, with some exceptions. The replicate naming in figure 3,4 , and 5 should be changed to be consistent with the tissue naming using in the manuscript. The right axes in figure 4 and 5 have several labels at each tick without explanation. I can guess what is meant by this, but it is non-standard and probably could be better represented by a table. The left axis shows % of genes, but it is unclear if this is % of DE genes or % of category that is DE. Figure 6 has several unlabeled components (see the numbers on the right, which I assume are # of genes but I am not sure). What are the values here? Means of clusters? Overall I did not find figure 3-5 and figure 7 especially useful to understand the manuscript. These figures simply report gene counts of various kinds that are perhaps better reported in tables.

Experimental design

The experimental question that the authors pose is relevant and meaningful, however the authors do not set the system up well to resolve this question. I am not convinced that there results tell us much about the development of different leaf types. Most of the sampling is done far after developmental specification is likely to have completed, however this is unknown to the review because the authors present no analysis of developmental timing, nor do the reference previous studies that have investigated this. An essential first step in developmental analysis is a clear understanding of the timing of particular events. Additionally, they do not deal with the fact that some samples are confounded with underwater sampling, which undoubtedly modifies the transcriptome considerably.
The description of bioinformatics analysis is overall incomplete, often not fully describing pipelines, software version numbers, and parameters. For example, Bowtie2, a sequence alignment program, is described as being used “to quantify transcript abundance”. This is followed by a “likelihood ratio test… to calculate gene expression levels and comparisons of the gene fragments per kb per million”. This is difficult to understand, and certainly incomplete with regards to what was actually done.
The assembly of the transcriptome is described slightly better, but I do not see any clear validation of its completeness.

Validity of the findings

I think that the production of a transcriptome is an important first step in the study of this species, but I find the manuscript’s focus on differentially expressed genes much less convincing. I do not think that there sampling clearly identifies relevant developmental time points that would allow them to identify important genes regulating development. As such I think the developmental framing of this manuscript is inappropriate. However, I do NOT think that this means that the comparisons are meaningless from a physiological standpoint, and perhaps the authors could re-envision the manuscript along those lines. Much of the manuscript is dedicated to describing potentially interesting, but cherry-picked genes from very large gene lists. Given potential complications with accurate annotations in this understudied system,
Additional brief, though not minor comments:
Why at the G clusters not consistent with the dendogram shown in the same figure?
Juvenile and adult are not appropriate terminology for a developing organ. Usually developmental stages are defined and then sampling is done based on those defined stages.
Terminology for id of DE genes is confusing here. I am not sure which threshold (or perhaps all thresholds were used ie pval threshold or fold change differences). Why do the authors report two fold change and log2 fold change great then one separately?
I am not sure what this sentence means “being annotated to 3,711 (8.73%) proteins of Elaeis”
Completeness of the transcriptome does not seem to have been tested.

---

## Round 0.2 · Minor Revisions

You need to address the further comments of the reviewers.

·

Basic reporting

no comment

Experimental design

I do not still see the data for the GO enrichment analysis. Figs. 4 and 5 show only the result of GO functional categorization.

Validity of the findings

no comment

Additional comments

Authors should use sample names throughout the article, instead of using library ID.

Reviewer 2 ·

Basic reporting

The authors have made an effort to satisfy my request for increased background information.
The authors have submitted their raw data to SRA, but this data has not been identified by sample appropriately. This study can not be replicated unless the specific samples are clearly described in the database.
Though the authors claim to have modified the figures (especially 3-5) I do not see these changes.

Experimental design

The experiment does not test for the transcriptional changes related to heterophyllus morphological change. I pointed this out in my previous review, but the authors chose to continue this focus. I think that the characterization of a new transcriptome is worthy of publication, but I remain unconvinced that the posed question is addressed. The authors could simply remove the developmental emphasis, point out that the transcriptome is a useful tool for gene identification, and that the species is interesting from a developmental standpoint. Besides the latter two points, I don't see how this study informs us about leaf morphological development.
If one wishes to understand the relationship between gene expression and morphological specification, then one has to compare gene expression patterns at stages where morphology is being specified. A focused description of the developmental process is necessary to accomplish this. The authors do not show that they have collected relevant time points, nor do they cite literature that indicates that their collection times are appropriate.
Alternatively the authors could focus on whatever developmental processes that are still ongoing at their collection times. These would have to be delineated.
The author's claim that tissue limitations make it impossible to study appropriate stages only means that they cannot address the stated question with this approach.
As I also stated in my previous review, the terms juvenile and adult are inappropriate to describe leaf development here. The authors may redefine these terms, but they are in conflict with already established terminology used throughout developmental studies. I strongly recommend that they use appropriate terminology, or simply identify leaves based on size.

Validity of the findings

See above.

---

## Round 0.3 · accepted · Accept

Thank you for addressing the reviewer comments.